# Precise measurements of chromatin diffusion dynamics by modeling using Gaussian processes

Guilherme M. Oliveira [1✉], Attila Oravecz[1], Dominique Kobi[1], Manon Maroquenne[1], Kerstin Bystricky [2], Tom Sexton [1✉] & Nacho Molina [1✉]

The spatiotemporal organization of chromatin influences many nuclear processes: from chromosome segregation to transcriptional regulation. To get a deeper understanding of these processes, it is essential to go beyond static viewpoints of chromosome structures, to accurately characterize chromatin's diffusion properties. We present GP-FBM: a computational framework based on Gaussian processes and fractional Brownian motion to extract diffusion properties from stochastic trajectories of labeled chromatin loci. GP-FBM uses higher-order temporal correlations present in the data, therefore, outperforming existing methods. Furthermore, GP-FBM allows to interpolate incomplete trajectories and account for substrate movement when two or more particles are present. Using our method, we show that average chromatin diffusion properties are surprisingly similar in interphase and mitosis in mouse embryonic stem cells. We observe surprising heterogeneity in local chromatin dynamics, correlating with potential regulatory activity. We also present GP-Tool, a user-friendly graphical interface to facilitate usage of GP-FBM by the research community.

[1] Institute of Genetics and Molecular and Cellular Biology (IGBMC) CNRS UMR7104, INSERM U1258, University of Strasbourg, Illkirch, France. [2] Molecular Cellular and Developmental Biology unit (MCD), Centre de Biologie Integrative (CBI) UPS, CNRS, Toulouse, France. ✉email: monteirg@igbmc.fr; sexton@igbmc.fr; molinan@igbmc.fr

The spatiotemporal organization of chromatin plays a crucial role in several nuclear processes: from cell division, where chromatin is compacted to facilitate chromosome segregation during mitosis, to gene regulation, where precise control of transcription correlates with specific long-range chromatin contacts[1]. Chromosome conformation capture techniques and imaging approaches have revealed fundamental structural features of chromatin at different resolutions. Of special interest are topological associated domains (TADs) which are characterized by an increased frequency of interactions between genomic loci within the same domain with reduced interactions across domains[2,3]. Remarkably, it has been shown that TAD organization can influence regulation of transcription[4] and that TADs are dismantled during mitosis where chromatin density dramatically increases[5,6]. However, much less is known about the diffusion properties of chromatin and how they depend on the genomic context. For instance, it is not clear whether or how transcriptional activation affects chromatin mobility, with previous studies giving seemingly conflicting results[7,8]. Insights into the dynamic properties of chromatin motion are required to understand how gene regulatory elements communicate within the nuclear space[9].

The simplest model to describe the diffusion of microscopic systems is Brownian motion, whereby movements are caused by random collisions of small particles within the system[10,11]. However, as a bulky polymer interacting with itself and the nuclear environment[12–14], chromatin displays sub-diffusive behavior, more constrained than classical Brownian motion for short periods of time[15–17]. Therefore, the mean squared displacement (MSD) of chromatin is expected to follow this relationship with time: $MSD \propto D_\alpha t^\alpha$. Two parameters thus describe the diffusion properties of chromatin: the apparent diffusion coefficient $D_\alpha$, indicating the "speed" of motion, and the anomalous coefficient $\alpha$, which for sub-diffusive behavior is <1 indicating greater constraint of movement. The traditional method used to estimate the diffusion parameters is based on calculating the MSD over time from measured trajectories and fitting the above theoretical expression to the data. More sophisticated methods based on particle displacement use higher-order moments[18] or probability density functions[19–21] (henceforth referred to as displacement distribution-based methods or DDB) to obtain more accurate estimations of the diffusion parameters. However, these methods do not use all the information contained in the trajectories as higher-order temporal correlations are discarded. Furthermore, errors due to measurement noise cannot easily be included into the analysis, and it is not possible to recover missing data points due to misdetection or occlusions.

We propose GP-FBM, a computational method based on Gaussian Processes (GP)[22,23] and fractional Brownian motion (FBM)[24–26], which improves and extends the concepts presented in[27,28]. Importantly, GP provides a consistent probabilistic framework that considers entire trajectories and thus utilizes all the available information. Trials on simulated data demonstrate a greater precision of GP-FBM in measuring diffusion parameters over MSD and DDB. Furthermore, as it is applied directly on trajectories, GP-FBM naturally takes into account localization errors and occlusions without the need to establish a fitted MSD curve or displacement distributions. We further extend this model to account for external sources of movement (e.g. displacement of whole nuclei or chromosomes) using underlying correlations between multiple trajectories, without the need to further develop substrate motion models and experiments for calibration[29]. Finally, we applied GP-FBM to two experimental systems to study chromatin diffusion properties in different contexts. First, we characterized chromatin dynamics in interphase and mitosis using tagged arrays inserted at random genomic locations in mouse embryonic stem (ES) cells. Although

chromatin density increases by a factor of three during mitosis[30], our results surprisingly indicate that there are no significant differences on average in the apparent diffusion or anomalous coefficients. Second, to compare the diffusion properties of different specific genomic regions, we performed double-labeling and live tracking experiments around the HoxA locus in mouse ES cells before and after induction of the genes with retinoic acid. We discover that, instead of having homogeneous diffusion properties across euchromatin, genomic loci significantly differ in both their apparent diffusion and anomalous coefficients. In some cases, altered chromatin diffusion properties correlate with underlying functions such as gene regulation or CTCF binding. The methods we have developed are integrated into a user-friendly package, GP-Tool, for use in the scientific community. Chromatin mobility has been overlooked in previous studies of genome functions, and we anticipate that GP-FBM will facilitate research in that area.

## Results

**Modeling diffusion dynamics with GP-FBM.** Traditional methods to analyze particle diffusion dynamics rely on particle displacements calculated between two frames at different time intervals, hence information on how precisely the particle moves between the two points is not considered. This has important drawbacks: higher-order temporal correlations within the trajectories are discarded; errors due to frame-dependent measurement noise cannot be easily included into the analysis; and missing data points due to misdetection or occlusions are ignored and cannot be recovered by inference. To address these problems we built a consistent probabilistic framework based on Gaussian Processes (GP)[22,23]. Briefly, a GP is defined as a collection of random variables such that every finite subset of them follows a multivariate normal distribution which is fully determined by its mean and kernel functions $\mu(t)$ and $\Sigma(t, t')$. We assume that a stochastic diffusion trajectory $x(t)$ of a given chromatin locus can be modeled as a Gaussian process with the following fractional Brownian kernel[24–26]:

$$\Sigma_{D_\alpha, \alpha}(t, t') = D_\alpha(|t|^\alpha + |t'|^\alpha - |t - t'|^\alpha), \quad (1)$$

where $D_\alpha$ is the apparent diffusion coefficient and $\alpha$ is the anomalous coefficient defined in the range $0 < \alpha < 2$. This kernel produces a generalized Brownian motion with a mean squared displacement $\langle r^2 \rangle = 2nD_\alpha t^\alpha$, where $n$ corresponds to the number of degrees of freedom. Notice that the traditional Brownian dynamics is recovered with $\alpha = 1$. Then the probability of observing a discrete trajectory $\mathbf{r} = \{r_i\}$ measured at a set of times $\mathbf{t} = \{t_i\}$ is given by the multivariate Gaussian distribution,

$$\mathcal{N}(\mathbf{r}|D_\alpha, \alpha, \boldsymbol{\mu}) \propto \exp\left[-\frac{1}{2}(\mathbf{r} - \boldsymbol{\mu})^T \Sigma^{-1}(\mathbf{r} - \boldsymbol{\mu})\right], \quad (2)$$

where the covariance matrix is defined as $\Sigma_{ij} \equiv \Sigma_{D_\alpha, \alpha}(t_i, t_j)$ and we take a constant $\boldsymbol{\mu}$ without loss of generality. Furthermore, we can easily incorporate localization errors by adding the diagonal term $\sigma_i^2 \delta_{ij}$ to the covariance matrix $\Sigma$, which assumes that errors are decorrelated and normally distributed with standard deviations $\boldsymbol{\sigma} = \{\sigma_i\}$ (see Methods). Ultimately, providing a trajectory $\mathbf{r}$ and the localization errors $\boldsymbol{\sigma}$, the likelihood (2) can be used to calculate estimates of the diffusion parameters $D_\alpha$ and $\alpha$ either via optimization or sampling using the Metropolis-Hastings algorithm[23,31].

We first test the performance of the GP-FBM method on synthetic trajectories simulated from a FBM model with a given time step ($dt$). To mimic the measurement noise observed in real trajectories, we introduce the localization error $\sigma$ and the occlusion rate $o$. An example of simulated trajectory and the effect of the measurement noise is shown in Fig. 1a. We then

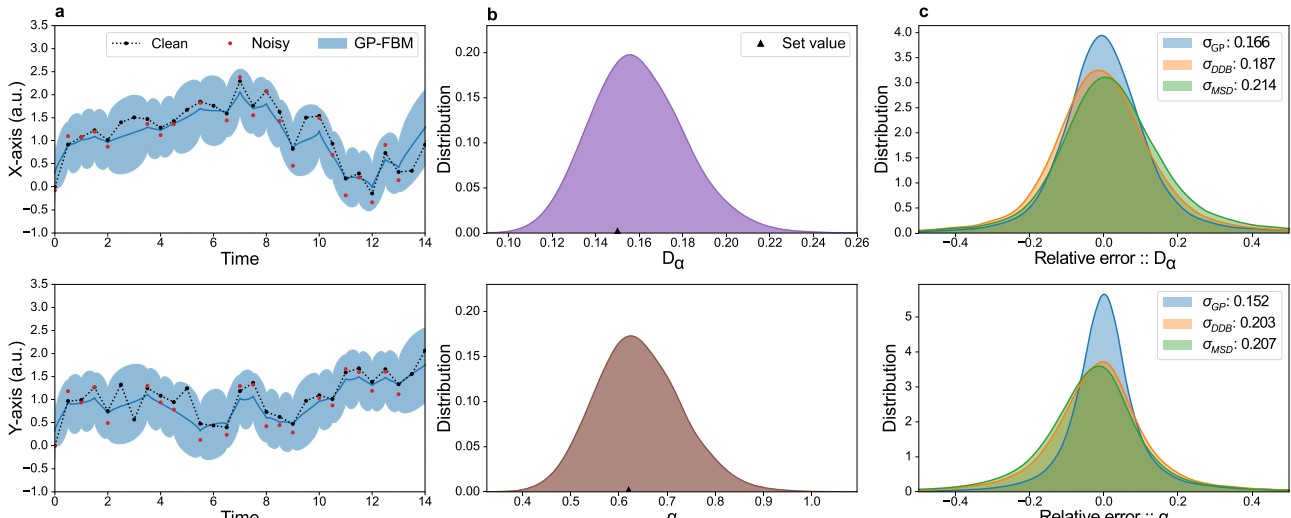

**Fig. 1 GP-FBM outperforms existing methods on simulated data. (a)** 2D trajectory sampled from a GP with the FBM kernel displayed in black. To match observed experimental trajectories in the following sections, Gaussian noise of 1/10 of a pixel is added to the positions and 10% of the points are removed according to a uniform random distribution. The noisy trajectory is displayed in red. The blue line shows the most probable trajectory with the blue shaded band representing the 95% credible interval as inferred by GP-FBM. **(b)** Posterior probability distributions for $D_\alpha$ and $\alpha$ inferred from trajectory in (a). Triangles denote the values set in the simulation. **(c)** Comparing inference relative errors as obtained by GP-FBM (blue), MSD (green) and DDB (orange) for 50000 simulated trajectories with uniform random parameters chosen in the range $0.01 < D_\alpha < 1.5$, $0.01 < \alpha < 1.9$, $0.1 < dt < 1.0$, $0.001 < \sigma < 0.25$ and $0 < o < 0.8$.

obtain a posterior distribution over the parameters $D_\alpha$ and $\alpha$ given the trajectory and the localization errors by combining the likelihood (2) with flat priors over all parameters, which can be sampled using Markov Chain Monte Carlo (MCMC) (see Fig. 1b and Methods). Interestingly, once the diffusion parameters are estimated, the power of the GP framework can be used to infer the most probable trajectory of the particle by removing measurement noise and predicting the particle position where occlusions or misdetections occurred (Fig. 1a). To systematically evaluate the performance of our method to infer diffusion parameters, we generated 50000 synthetic trajectories using uniformly distributed random values of $D_\alpha$ and $\alpha$ in the range $0 < D_\alpha < 1.5$ and $0 < \alpha < 2$. For generality, we also sample the simulation time step ($dt$), localization error ($\sigma$) and occlusion ratio ($o$) from a uniform random distribution in the respective ranges: $0.1 < dt < 1.0$, $0.001 < \sigma < 0.25$ and $0 < o < 0.8$. We compared the results obtained using our approach on the simulated data with the traditional MSD and DDB methods. GP-FBM clearly outperforms both methods, producing smaller relative errors of parameter estimation over all (Fig. 1c) and across different parameter ranges (Supplementary Figs. 1 and 2). Unlike GP-FBM, both MSD and DDB methods require trajectories to be split into individual displacements, thus neglecting higher-order temporal correlations that the trajectories may contain. Therefore GP-FBM method can optimally infer diffusion parameters from single trajectories, using all the information contained in the data and thus achieving greater precision.

**Accounting for substrate movement with GP-FBM.** Often a particle may be subject to secondary movement that is entangled with its diffusion dynamics. In chromatin dynamics, this movement is frequently associated with the substrate in which the particle is diffusing, such as cell displacement, membrane fluctuations or chromatin reallocation, as well as technical considerations such as thermal drift and undesired media flow. If overlooked, this may result in over-estimation of the diffusive properties. However, when two or more particles are measured in the same context, this substrate movement can be accounted for

by analyzing the cross-correlation introduced between the particle trajectories. To that end, we developed a covariance model that takes advantage of the GP-FBM framework to quantify substrate movement and handle the cross-correlation that it may introduce into the movement of all particles (see Methods). In the case of two particles, we obtain the probability distribution,

$$\rho(\boldsymbol{r_1}, \boldsymbol{r_2}|\boldsymbol{\alpha}, \boldsymbol{D_\alpha}) \propto \exp\left\{ -\frac{1}{2} \begin{pmatrix} \boldsymbol{r_1} \\ \boldsymbol{r_2} \end{pmatrix}^T \begin{pmatrix} \Sigma_1 + \Sigma_R & \Sigma_R \\ \Sigma_R & \Sigma_2 + \Sigma_R \end{pmatrix}^{-1} \begin{pmatrix} \boldsymbol{r_1} \\ \boldsymbol{r_2} \end{pmatrix} \right\},$$
(3)

where $\Sigma_1$, $\Sigma_2$ and $\Sigma_R$ are FBM covariance matrices for the two particles and substrate respectively with diffusion parameters $\boldsymbol{D_\alpha} = \{D_{\alpha,1}, D_{\alpha,2}, D_{\alpha,R}\}$ and $\boldsymbol{\alpha} = \{\alpha_1, \alpha_2, \alpha_R\}$. This method can easily be extended for higher number of particles, limited only by required computational power in practice, even though most of the correction is already achieved with two particles (Supplementary Fig. 3). In this study, we restrict our analysis to five particles per cell.

To demonstrate the utility of this approach, we generated 2000 synthetic trajectories as before, but now including substrate movement, generating vectors $\boldsymbol{r_i}$ as a combination of substrate displacement $\boldsymbol{R}$ and the actual particle displacement $\boldsymbol{a_i}$ (Fig. 2a). Simulations are generated with 10% occlusion rate and localization error, which are values commonly found in our experiments. As expected, $D_\alpha$ and $\alpha$ tend to be overestimated if substrate movement is unconsidered; however, the parameters are more precisely determined when the substrate correction is incorporated into the model (Fig. 2b,c). The method is also able to estimate the dynamic properties of the substrate and, albeit with less precision, the substrate movement itself (see Fig. 2d, Fig. S4 and Methods). Furthermore, we tested the performance of the method depending on the number of tracked particles subjected to the same substrate movement. Precision is increased with use of more particles, but the bulk of the error is already removed with only two particles (see Supplementary Fig. 3). Finally, we showed that GP-FBM outperforms the DDB method even when the substrate movement is taken into account (see Methods and Supplementary Fig. 5). In conclusion, GP-FBM has the ability to

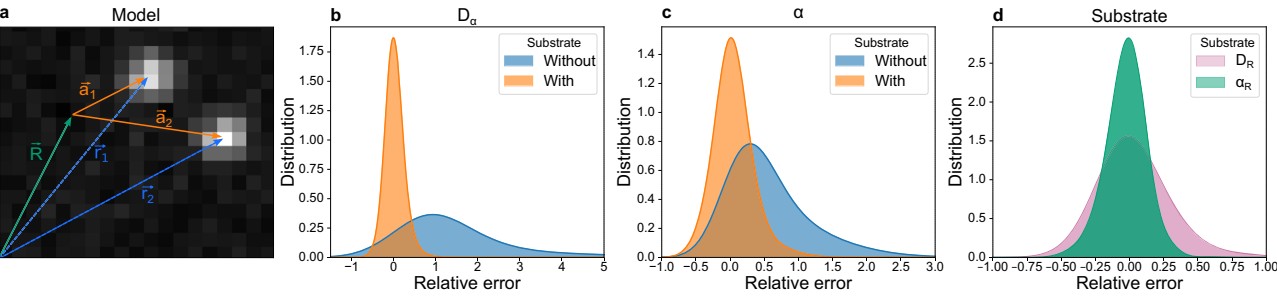

**Fig. 2 GP-FBM can correct for substrate movement to improve estimation of diffusion parameters.** (**a**) Scheme showing the measured trajectories of particles $r_1$ and $r_2$ as the combination of substrate motion $R$ and particles' diffusion with respect to the substrate, $a_1$ and $a_2$. (**b, c**) Distributions of relative errors for the estimation of diffusion parameters on 2000 pairs of simulated trajectories with uniform random parameters ($0.1 < D_i < 1.1$, $0.3 < \alpha < 1$, $0.1 < D_R < 1.1$ and $0.7 < \alpha_R < 1.7$; $dt = 0.5$, $\sigma = 0.1$ and $o = 0.1$) when substrate movement is accounted for (orange) or not (blue). (d) Distributions of relative errors for the estimation of diffusion parameters for the substrate.

remove the substrate movement from the analysis which is demonstrably important for precise measurement of diffusion parameters. Other approaches can be applied to estimate cell movements and correct trajectories[32,33], but GP-FBM has the advantage of being able to derive this information directly from the trajectories themselves, provided that two or more particles are tracked per cell. Consequently, we are able to automatically characterize the substrate movement and remove it from the analysis without the need of extra image processing steps thanks to the cross-correlation that this external movement imprints in the particle diffusion dynamics.

**Analyzing chromatin dynamics in interphase and mitosis.** Due to chromosome compaction and condensation, chromatin density increases by a factor of three during mitosis[30]. Although the structure of mitotic chromatin has been intensively studied[5,34,35], it is unknown if or how the higher density and rearrangement of chromatin fibers affects chromatin diffusion properties. To measure chromatin dynamics in interphase and mitosis, we used a mouse ES cell line carrying approximately 20 *TetO* arrays of 7 kb length inserted at random genomic locations[36]. GFP::TetR is stably expressed in these cells, where it binds to the *TetO* arrays for the simultaneous visualization of several chromatin loci in each cell. We performed confocal live-imaging and distinguished interphase and mitotic cells by DNA staining using Hoechst 33342, recording images at 4 frames per second for 75 s. To increase the number of mitotic cells, we also performed live-imaging experiments on cells arrested in prometaphase with nocodazole (see Fig. 3a and Methods). We tracked spots using ICY[37] and enhanced particle localization precision by fitting a 2D Gaussian function to the signal of the tracked spots (see Methods and Supplementary Fig. 6). Before applying the GP-FBM probabilistic framework, we first determined whether the measured stochastic trajectories present, to a certain approximation, self-similar Gaussian distributed displacements and a FBM velocity autocorrelation function (see Methods). Interestingly, that seems to be the case for chromatin movements at the time scale of this study, hence GP-FBM is an appropriate approach for the analysis (Supplementary Figs. 7 and 8).

Comparing the performance of GP-FBM with and without substrate movement correction, it was apparent that actively dividing mitotic cells had greater substrate movement (presumably due to coordinated alignment and movement of chromosomes by the mitotic spindle), but that appreciable correction was required for precise chromatin dynamics measurements in all conditions (Fig. 3b). Surprisingly, we observed no significant differences in the mean apparent diffusion or the mean anomalous coefficients between interphase and mitotic chromosomes ($p \geq 0.05$), suggesting that condensation may not

necessarily affect the average local diffusion dynamics of chromatin (Fig. 3c and d). We observed a small but significant increase in the anomalous coefficient of mitotic-arrested cells compared to interphase, which might be related to the effect that nocodazole has on microtubule formation and thus mitotic chromosome stability[38].

Interestingly, we obtained a wide range of estimated $D_\alpha$ and $\alpha$ coefficients indicating a remarkable spot-to-spot variability in their diffusion dynamics, even when correcting for substrate movement. This variability could partially be caused by differences in the state of the analyzed cells (inter-cell variability) leading to different overall chromatin dynamics. Alternatively, differences in the chromatin context of the genomic loci could lead to specific diffusion dynamics (intra-cell variability). Applying the law of total variance (see Methods), we quantified the contribution of inter-cell vs intra-cell variability (Fig. 3e and f). Strikingly, as much as 75% of the variability in $D_\alpha$ and 65% in $\alpha$ could be explained by differences within the same cells. This estimate is even higher when substrate movement is taken into account, especially in the case of mitotic cells, when mouse ES cells tend to detach from their colonies, becoming more prone to movement. In contrast, the nocodazole arrested cells are allowed to sediment onto the glass surface, thus are less mobile during imaging. Together, this suggests that different genomic loci may have characteristic local diffusion properties due to their specific chromatin or nuclear context.

**Distinguishing locus- and cell-specific diffusion properties.** Except for a tendency for chromatin mobility to be reduced at centromeric or telomeric locations in yeast[39], little is known about how different genomic contexts may affect dynamics of the underlying chromatin. Further, previous studies give conflicting views on whether transcriptional activation can increase local confinement of a gene (as observed in the same cell before and after estrogen stimulation[7]) and/or increase gene mobility (as observed comparing cells before and after differentiation[8]). To compare the diffusion properties of different specific genomic regions, we performed double-labeling and live tracking experiments with the ANCHOR system[40] around the HoxA locus in mouse ES cells before and after induction of Hox genes with retinoic acid. We engineered the ANCH1 and ANCH3[7] labels into different locations within the same allele to generate two ES lines with equidistant probes assessing inter-TAD (T1-T2) or intra-TAD (T2-T3) associations (Fig. 4a, b) and imaged at 2 frames per second for 2 min. As may be expected, the average inter-probe distance was higher for the inter-TAD than intra-TAD combination, but with large heterogeneity in the distance distributions (Fig. 4c;[41]). Interestingly, Hox gene induction had no effect on intra-TAD distances within the neighboring domain,

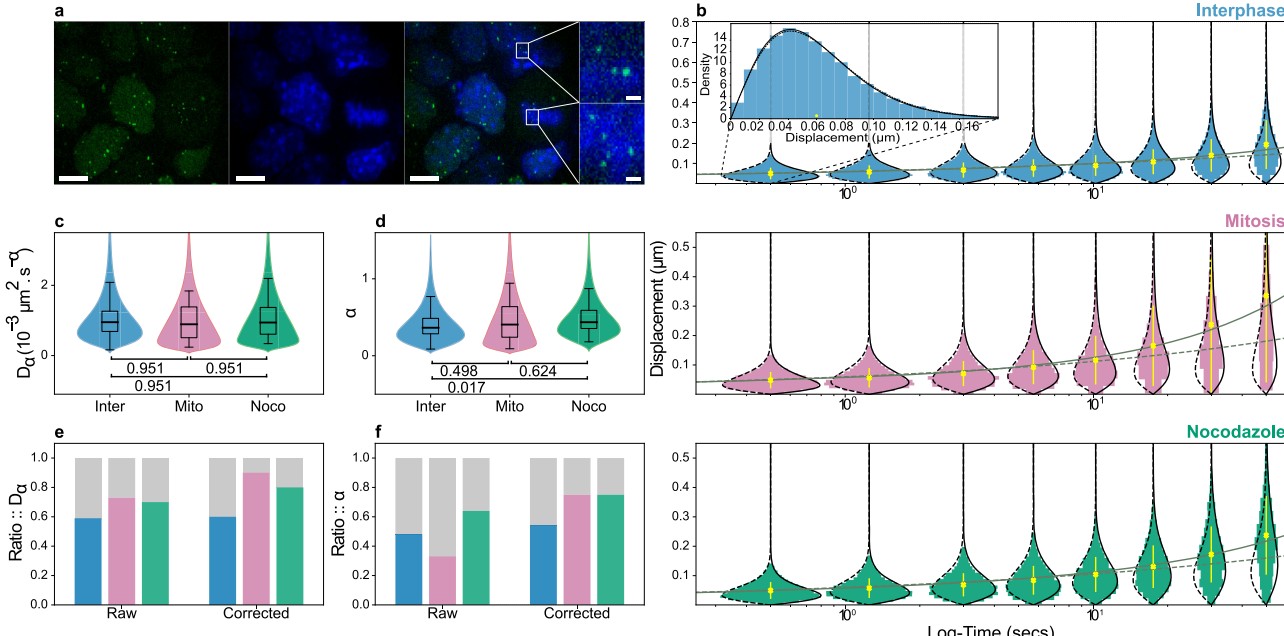

**Fig. 3 Average chromatin dynamics is similar in interphase and mitosis, but are highly variable across loci. a** Maximum projection images of ES cells containing spots of TetR::GFP (green) bound to *TetO* arrays, with DNA (blue) stained with Hoechst (scale bar is 10 μm). Inset shows magnification of the selected area (scale bar 1 μm). **b** Histograms showing the displacement distributions of loci in interphase (blue), mitosis (red), or mitotic arrest (green), plotted for different time points. Theoretical distributions with and without substrate correction are shown with solid to the right and dashed black lines to the left, respectively. Inset shows a rotated displacement distribution for clarification purpose. The observed mean displacement with standard deviation is shown in yellow and theoretical curves with and without substrate correction are shown in gray continuous and dashed lines, respectively. The model with substrate movement fits better the data, suggesting a great substrate effect, especially for active mitotic cells at greater time intervals. **c**, **d** Distribution of estimated $D_\alpha$ and $\alpha$ in the three conditions, correcting for substrate movement. Boxes represent the interquartile range and whiskers 95% of the data. Medians are shown as solid lines inside the boxes. **e**, **f** Estimations of the inter-(gray) and intra-cell (color) proportions of total variance for $D_\alpha$ and $\alpha$ in interphase (blue), mitosis (red), and mitotic arrest (green). The number of spots analyzed in interphase, mitosis, and nocodazole tratement were $n = 249$, $n = 23$, and $n = 36$, respectively, over a total of 3 independent experiments. Source data are provided as a Source Data file.

but increased inter-TAD distances, supporting the idea of general TAD reinforcement as cell differentiation is induced[42]. As tests of Gaussianity and velocity autocorrelation again verified approximation of chromatin dynamics to FBM (see Methods and Supplementary Figs. 7 and 8), we performed GP-FBM for the three loci and found that, in undifferentiated ES cells, although they have equivalent apparent diffusion rates, region T1 is significantly more constrained than T2 or T3 (Fig. 4d, e). Closer inspection of the ES (and differentiated neuronal precursor cell) epigenomic profiles around these regions showed that T1 is close (<15 kb) to a putative active enhancer of *Halr1* (Supplementary Fig. 9). This gene encodes the long non-coding RNA *Haunt*, whose specific expression in ES cells is linked to suppression of the HoxA genes[43]. Active histone modifications around T1, compared to the silent T2 and T3 regions, correlates with a greater constraint of the chromatin, in line with a previous study of an estrogen-induced gene[7] and predictive polymer models[14]. Hox gene induction by retinoic acid had no significant effect on the diffusive rate of T1 but did reduce locus constraint (Fig. 4d, e). In contrast, the region T2, which lacks any known epigenomic or regulatory features, had increases in $D_\alpha$ and $\alpha$, perhaps indicative of general chromatin remodeling caused on onset of differentiation. Curiously, T3 became more constrained on retinoic acid treatment, with a concomitant increase in mobility. This region contains sites bound by the architectural protein CTCF, whose binding is either lost or reduced on differentiation to neuronal precursors (Supplementary Fig. 9). CTCF is proposed to form a roadblock for cohesin-mediated loop extrusion processes[44,45], and this may be expected to play out in alterations to local chromatin dynamics, although this has been largely unexplored.

Overall, these results show previously unappreciated locus-specific variation in chromatin diffusive properties, which in some cases correlate with underlying histone modifications or CTCF binding.

**GP-Tool allows user-friendly application of GP-FBM**. To facilitate use of GP-FBM by the community, we developed a freely available graphical user interface called GP-Tool (Fig. 5; github.com/guilmont). GP-Tool contains 4 plugins: movie, alignment, trajectories and g-process. The movie plugin allows the user to open TIFF files, display basic ImageJ and OME metadata, define colormaps for each channel and manually correct for contrast. The alignment plugin runs the algorithm described in Methods to digitally correct chromatin aberration and possible camera alignment issues. Alternatively, the user can manually modify each of the parameters. Finally, the g-process plugin allows to infer optimal values for the apparent diffusion and anomalous coefficients for several cells in the same movie whilst correcting for substrate movement if two or more particles are selected. It is also possible to use a Metropolis-Hastings sampler to obtain the posterior probability distribution associated with each of these parameters. Once the analysis is complete, the tool provides the possibility to save the results into JSON files. It also provides export functions to save tables in CSV format. All these formats are easily parsed in all major computing languages, such as C/C++, Python and R. Finally, GP-Tool provides shared libraries and C/C++ examples for batching multiple movies. A complete documentation of the software can be found in the aforementioned Github account and in Supplementary Materials.

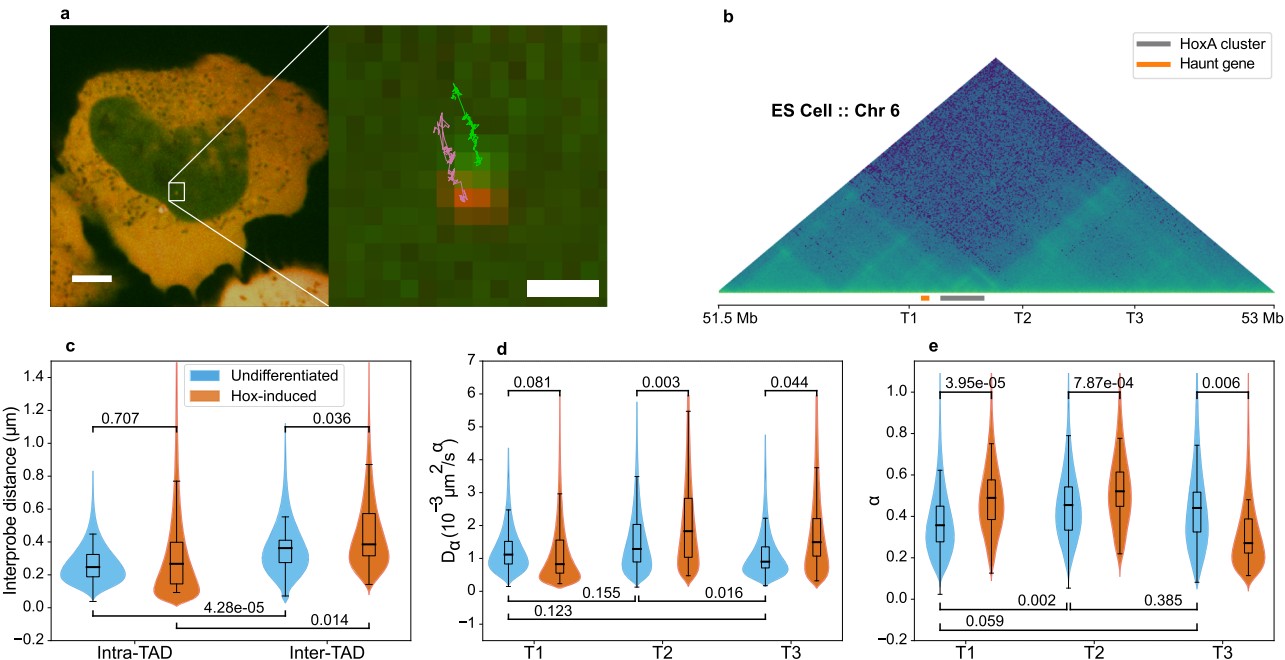

**Fig. 4 Chromatin dynamics of three chromatin loci within the HoxA genomic region. a** Example of microscopy image obtained for ANCHOR cell lines (scale 5 μm) with tracked trajectories shown in amplified subpanel (scale 0.44 μm). **b** ES Hi-C map around the HoxA cluster, illustrating the locus TAD structure. The positions of ANCHOR probes (T1, T2, and T3), the *Haunt* gene, and the HoxA cluster are shown underneath the map. **c** Inter-probe distances measured for ES cells before (blue) and after (red) treatment with retinoic acid. **d, e** Comparisons of apparent diffusion and anomalous coefficients between the three labeled loci, before (blue) and after (red) treatment with retinoic acid. Boxes represent the interquartile range and whiskers 95% of the data. Medians are shown as solid lines inside the boxes. A total of $n = 56$ and $n = 67$ spots were analyzed from the inter-TAD (T1-T2) cell line before and after treatment, respectively, from 3 independent experiments; $n = 51$ and $n = 13$ points were used for intra-TAD (T2-T3) cell line before and after differentiation, respectively, from 2 independent experiments. Source data are provided as a Source Data file.

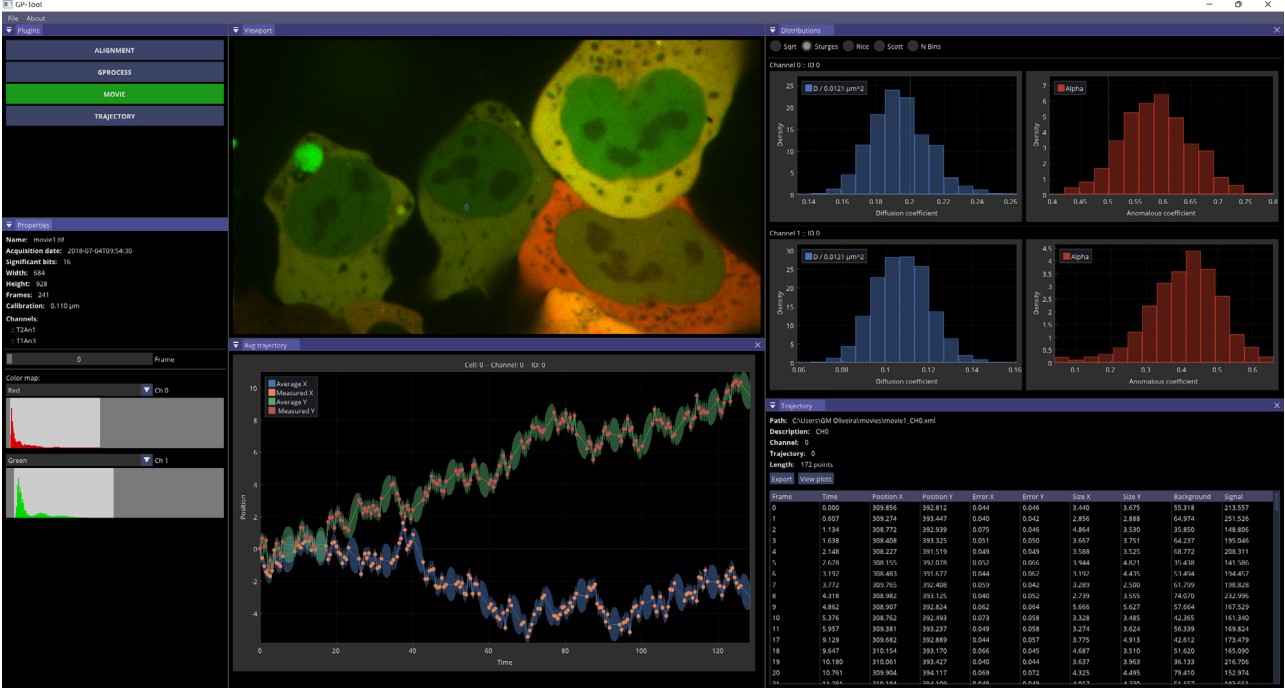

**Fig. 5 GP-Tool: A graphical user interphase to apply GP-FBM on microscopy movies.** Viewport displays movie under analysis. The distributions are estimations of the posterior distribution for $D_\alpha$ and $\alpha$ via Metropolis-Hasting sampling. Trajectory tab displays detections with enhanced localization and other important parameters such as estimated localization error, particle size and signal. The average trajectory tab displays interpolated curves with the most probable path taken by the particle with a 95% credible interval.

## Discussion

We developed GP-FBM, a Bayesian framework that combines the inference power of Gaussian processes with fractional Brownian motion, a flexible model to describe Gaussian-like diffusion dynamics. Importantly, chromatin loci show Gaussian and self-similar displacement distributions, indicating that FBM is an adequate approximation to assessing chromatin movement, at least for the time scales over which experiments are commonly performed (from a few up to hundreds of seconds). Notice that for longer time scales a crossover has been observed between different diffusion regimes and our model would have to be modified to incorporate this behavior[16]. Note also that a myriad of other biological systems have non-Gaussian dynamics, so would not be suitably analyzed by GP-FBM[20,21,46,47].

GP-FBM treats stochastic trajectories as a whole without pre-processing or extracting limited statistics from them. Therefore, this approach utilizes optimally all the information contained in the data by incorporating higher-order temporal correlations into the analysis. In addition, the Gaussian process framework allows easy integration of spot-dependent localization errors which translate into a consistent weighting of time points, depending on the precision at which the spot position is determined. Furthermore, missing data due to spot misdetection or occlusion does not hinder the analysis and, on the contrary, GP-FBM can be used to probabilistically assign spot positions for any given time point. Finally, when two or more particles are tracked in similar context, GP-FBM uses possible cross-correlations between trajectories to characterize substrate movement and, therefore, remove it from the analysis. A number of other methods have been developed over recent years to better characterize diffusion of particles, employing variations of MSD[18,48], probability density functions for particle displacements[19–21], Bayesian inference[27,28] or even machine learning approaches[49,50]. However, to our knowledge, these methods are either not readily applicable to experimental data, require extra experiments to precisely measure complementary parameters to determine background movement, require large amounts of varying training sets to account for different shapes/types of input data, and/or are not robust to mislocalization and occlusion events that are commonplace in imaging experiments. Benchmarking against MSD and DDB, GP-FBM show improved results over all combinations and ranges of tested parameters. GP-FBM is thus a precise and robust tool. Providing that the model assumptions are fulfilled, this increase in accuracy can be crucial to study changes in diffusion dynamic properties in different conditions.

We applied GP-FBM to two ES cell systems and observed a large variability in chromatin dynamics when comparing individual cells and comparing different loci. A fraction of this variability can be explained by differences across cells, especially in interphase cells, indicating that cell state (cell cycle or metabolism) may globally influence chromatin dynamics. However, the majority of the observed variability is related to differences across loci. Unexpectedly, chromatin exhibits similar average diffusion dynamics in interphase and mitosis despite a large difference in chromatin density. This result may be related with recent findings showing that mitotic chromatin is not as inaccessible and inert as previously thought. Indeed, several studies have shown that mitotic chromatin is bound by transcription factors[51,52] and some genes are even transcribed during mitosis[53]. In contrast, different genomic loci can have striking differences from average chromatin dynamic properties, which in some cases correlate with underlying functional chromatin marks. It has been previously proposed that chromatin mobility is affected directly by transcription, although results were seemingly conflicting[7,8]. More widespread application of GP-FBM to labeled transcribed loci and other specific regulatory elements, such as enhancers or TAD borders, will likely uncover more interesting functional links between genome function and the dynamics of its component chromatin.

Finally, we present GP-Tool a graphical user interface that helps to perform GP-FBM analysis on microscopy movies with only a few mouse clicks. Importantly, this tool and the GP-FBM framework can be applied to study not only chromatin dynamics but potentially any labeled particle that can be tracked over time providing that Gaussianity and other assumptions of FBM are met. Alternatively, the FBM kernel used in this study can potentially be replaced by alternative kernels that may better describe dynamics of other systems. We thus anticipate that GP-FBM and GP-Tool will greatly facilitate the analysis of diffusion dynamics in biology.

## Methods

### Cell lines, culture, and treatments

*Transgenic TetO ES line.* The mouse ES cell line was kindly provided by Dr. Luca Giorgetti. It is derived from an X0 clone of the PGKT2 subclone of the feeder-independent PGK12.1 mouse ESC line which was engineered by co-transfection with pBROAD3-TetR-ICP22NLS-eGFP and pcDNA3.1Hygro to stably express the TetR-eGFP recombinant protein after random integration and hygromycin selection (250 μg/ml) as described in ref. [36,54]. The piggyBac transposon system was then used to generate cells with 20–25 stable random integrations of a 150 TetO binding site array as described in[36]. Cells were cultured on 0.1% gelatin-coated culture plates in DMEM (4.5 g/l glucose) supplemented with GLUTAMAX-I, 15% fetal calf serum (ES cell culture tested), 0.1 mM beta-mercaptoethanol, 1,500 U/ml leukemia inhibitory factor (LIF; produced in house), and 0.1 mM non-essential amino acids in 5% $CO_2$ at 37 °C. Mitotic arrest was performed by treating the cells for 5 h with 100 ng/ml Nocodazole (Sigma, M1404-2MG). This cell line is available upon reasonable request.

*Transgenic ANCHOR ES lines.* J1 mouse ES cells were grown on gamma-irradiated mouse embryonic fibroblast cells under standard conditions (4.5 g/L glucose-DMEN, 15% FCS, 0.1 mM non-essential amino acids, 0.1 mM beta-mercaptoethanol, 1 mM glutamine, 500 U/mL LIF, gentamicin), then passaged onto feeder-free 0.2% gelatin-coated plates for at least two passages to remove feeder cells before subsequent transfections. The two ("inter-TAD" and "intra-TAD") ANCHOR transgenic lines were generated by sequential CRISPR/Cas9-mediated knock-in experiments in the following manner. First, flanking homology arms (mm9 chr6: 52,320,061-52,321,144, and chr6: 52,321,145-52,322,244) were introduced by PCR amplification and Gibson assembly into a vector containing ANCH1 sequence[40]. This vector (1 μg) was co-transfected with 3 μg of a vector containing Cas9-GFP, a puromycin resistance marker, and the scaffold to transcribe the sgRNA specific to the T2 insertion site (CGGCGCGCACTTAA-CACCAA; vector generated by the IGBMC Molecular Biology platform) in 1 million cells with Lipofectamine-2000. Two days after transfection, the cells were cultured for 24 h with 3 μg/ml puromycin, then 48 h with 1 μg/ml puromycin to enrich for transfected cells, before sorting individual GFP-positive cells on to feeders to amplify individual clones. Clones with the correct sequence were screened by PCR and sequencing, then the CRISPR knock-in process was repeated to insert the ANCH3 sequence[7] into either the T1 site ("inter-TAD" line; homology arms at chr6: 52,013,471-52,014,370 and chr6: 52,014,371-52,015,270; gRNA sequence AATCGAGCTCACGCCATTAG) or the T3 site ("intra-TAD" line; homology arms at chr6: 52,622,955-52,623,855 and chr6: 52,623,856-52,624,755; gRNA sequence TATGCTGAGGCGTGTCGCAA). Final clones were verified for maintained pluripotency by qRT-PCR to assess Oct4, Nanog (e.g. Supplementary Fig. 9), and Sox2 expression. Subsequent microscopy experiments (see below) confirmed heterozygous incorporation of the ANCH sequences (detection of one specific spot per ANCH sequence per cell) within the same allele (two spots were always in close proximity). This cell line is available upon reasonable request.

*OR transfection.* 150,000 cells are plated two days prior to imaging off feeder cells onto laminin-511-coated 35 mm glass bottom petri dishes, and transfected with 3 μg OR1-EGFP and 3 μg OR3-IRFP plasmids (vectors available from NeoVirTech (contact@neovirtech.com); were modified from original source by changing the C-terminal fluorescent protein sequence, introducing Kozak sequence before the translation start site and replacing the CMV promoter with EF-1α) using Lipofectamine-2000. After two days, the medium is changed to remove dead cells, before passing directly to microscopy.

*Hox induction.* ES cells were passaged without feeders and cultured on laminin-511 for two days without LIF, then for a subsequent three days without LIF and with the addition of 5 μM retinoic acid. One day after the addition of retinoic acid, the cells are transfected with the OR proteins as previously.

## Microscopy

*Live cell imaging of TetO ES cells.* 35 mm glass-bottom dishes (Ibidi 81158) were coated with 10 μg/ml fibronectin human plasma (Sigma, F2006-1MG) in PBS for 45 min at room temperature. A total of $3–5 \times 10^5$ cells were seeded one day before imaging, then the medium was replaced by phenol-red-free medium containing 500 ng/ml Hoechst 33342 (Invitrogen, H3570). Cells arrested in mitosis were collected on the day of imaging by "shake-off", incubated with 0.25% Trypsin-1 mM EDTA (Invitrogen, 25200-072) for 1 min at 37 °C and washed, and placed on fibronectin-coated glass-bottom dishes in phenol-red-free medium containing 100 ng/ml Nocodazole and 500 ng/ml Hoechst 33342. Confocal live-cell imaging was performed on a Nikon Eclipse Ti-E inverted widefield microscope (Perfect Focus System) equipped with a CSU-X1 confocal scanner unit and an Evolve back-illuminated EMCCD camera (Photometrics). Images were recorded using $100 \times$ HC Plan APO oil immersion objective (Leica, NA 1.4). Intensities were set to 10% for the 405 nm and 30% or 50% for the 491 nm lasers, with exposure times of 100 ms and 50 ms or 25 ms, respectively. 5 z-stacks with 0.5 μm distances were recorded for each channel. 301 time-lapse images were recorded only in the 491 channel.

*Live cell imaging of ANCHOR ES cells.* Imaging experiments were performed on an inverted Nikon Eclipse Ti microscope equipped with a PFS (perfect focus system), a Yokogawa CSU-X1 confocal spinning disk unit, two sCMOS Photometrics Prime 95B cameras for simultaneous dual acquisition to provide 95% quantum efficiency at 11 μm × 11 μm pixels and a Leica 100× oil objective (HC PL APO 1,4 oil immersion). We excited EGFP and IRFP with a 491 nm (100 mw) and a 635-nm laser (> 28 mW), respectively. We detected green and far red fluorescence with an emission filter using a 525/50 nm and a 708/75 nm detection window, respectively. A thermostated heater (Tokai Hit Stage Top Incubator) allowed for heating at 37 °C, humidity, and $CO_2$ control (5%). Time-lapse analysis of GFP and IRFP foci was performed in 2D acquiring 241 time points at a 0.5 s time interval. The system was controlled using Metamorph 7.10 software. Time-lapse was concatenated into single TIFF file.

## RT-qPCR

RNA was extracted from cells using the Nucleospin RNA extraction kit (Machery-Nagel), then cDNA was prepared with SuperScript IV (Invitrogen), following the manufacturer's instructions and using random hexanucleotides as primers. The cDNA was quantified by qPCR on a LC480 LightCycler (Roche), using QuantitTect SYBR Green PCR kit (Qiagen). Amplification was normalized to GAPDH. Primer sequences are given in Supplementary Table 1.

## Image pre-processing

*Spot detection and tracking.* Spot detection and tracking for all movies was performed with ICY, an image analysis software[37]. Localization precision was then enhanced by assuming that the spots have the shape of a 2D Gaussian function as follows,

$$S_{x,y} = I_o \exp\left\{ -\frac{1}{2} \begin{pmatrix} x - \mu_x \\ y - \mu_y \end{pmatrix}^T \begin{bmatrix} L_x^2 & \theta L_x L_y \\ \theta L_x L_y & L_y^2 \end{bmatrix}^{-1} \begin{pmatrix} x - \mu_x \\ y - \mu_y \end{pmatrix} \right\} + B_G. \quad (4)$$

with $\mu_i$ representing the center of mass of the spot, $L_i$ its size in directions x and y, $-1 < \theta < 1$ a possible rotation, while $B_G$ and $I_o$ are background and spot signal, respectively. We optimize its localization using the NM-Simplex method[55] and estimate localization error using the Metropolis-Hastings algorithm[23,31]. This method is implemented and automatically runs when trajectories are loaded in GP-Tool. For more information see Supplementary Figs. 6 and 10.

*Multi-channel alignment correction.* For the ANCHOR ES cell line experiments, we used a spinning disk microscope setup with 2 cameras, i.e., one per channel. Even though these cameras were aligned manually using fluorescent beads, we could still observe non-negligible differences between images captured in both cameras. Furthermore, even in rare situations when both cameras were properly aligned, we could observe effects of chromatic aberrations towards the edges of the image due the different wavelengths used. To correct for such problems, we performed digital post-alignment using a generic set of affine transformations including translation, rotation and scaling as defined in

$$\Omega = \begin{pmatrix} s_x & 0 & (1-s_x)W/2 \\ 0 & s_y & (1-s_y)H/2 \\ 0 & 0 & 1 \end{pmatrix} \begin{pmatrix} 1 & 0 & d_x+c_x \\ 0 & 1 & d_y+c_y \\ 0 & 0 & 1 \end{pmatrix} \begin{pmatrix} \cos(\theta) & \sin(\theta) & 0 \\ -\sin(\theta) & \cos(\theta) & 0 \\ 0 & 0 & 1 \end{pmatrix} \begin{pmatrix} 1 & 0 & -c_x \\ 0 & 1 & -c_y \\ 0 & 0 & 1 \end{pmatrix}, \quad (5)$$

where, $s_i$ accounts for scaling in directions x and y, $d_i$ accounts for translation in both directions and $\theta$ is the angle of rotation between both channels in relation to point $c_i$.

To infer optimal parameters for correction, we used 5 frames from all the movies recorded in the session and maximize the following likelihood using the Nelder-Mead simplex method[55]

$$\log P \propto -\frac{WH}{2} \log \left\{ \sum_{k,l} \left[ I_2(k,l|\Omega) - I_1(k,l|\mathbb{1}) \right]^2 \right\}, \quad (6)$$

where W and H correspond to width and height of images and $I_r(k,l|A)$ is the value of pixel (k,l) in channel r given transformation A. Here, $\mathbb{1}$ represents the identity matrix. Supplementary Fig. 11 shows examples of misaligned images and how the alignment improves greatly after applying our algorithm.

## Derivation of GP-FBM models

*Fractional Brownian motion.* The covariance function of FBM can easily be derived from the assumption of two basic properties[56]: stationary increments $B(t) - B(s) \propto B(t-s)$ and a power-law variance, $\langle B(t)^2 \rangle \propto |t|^\alpha$. Then, the off-diagonal terms of the covariance function can be determined as follows:

$$\langle B_\alpha(t)B_\alpha(s) \rangle \propto \frac{1}{2} \langle [B_\alpha(s) - B_\alpha(s) + B_\alpha(t)]B_\alpha(s) + B_\alpha(t)[B_\alpha(t) - B_\alpha(t) + B_\alpha(s)] \rangle$$
$$= \frac{1}{2} \langle B_\alpha(s)^2 + B_\alpha(t-s)B_\alpha(s) - B_\alpha(t)B_\alpha(t-s) + B_\alpha(t)^2 \rangle$$
$$= \frac{1}{2} \langle B_\alpha(t)^2 + B_\alpha(s)^2 + B_\alpha(t-s)(B_\alpha(s) - B_\alpha(t)) \rangle$$
$$= \frac{1}{2} \langle B_\alpha(t)^2 + B_\alpha(s)^2 - B_\alpha(t-s)^2 \rangle$$
$$= \frac{1}{2} (|t|^\alpha + |s|^\alpha - |t-s|^\alpha). \quad (7)$$

Finally, the apparent diffusion coefficient $D_\alpha$ is introduced as a proportionality factor to re-scale mobility, leading to the final kernel as presented in the main text:

$$\Sigma_{D_\alpha,\alpha}(t,s) = 2D_\alpha \langle B_\alpha(t)B_\alpha(s) \rangle. \quad (8)$$

We can also calculate the velocity autocorrelation function for the FBM model[26], which can be easily calculated from experimental trajectories using

$$C_\nu^{(\epsilon)}(\tau) = \frac{1}{\epsilon^2} \langle (x(\tau+\epsilon) - x(\tau))(x(\epsilon) - x(0)) \rangle, \quad (9)$$

where velocity is defined as $\nu(\tau) = \epsilon^{-1}[x(\tau+\epsilon) - x(\tau)]$. Using that, the theoretical curve for velocity autocorrelation function for FBM is calculated to be

$$\frac{C_\nu^{(\epsilon)}(\tau)}{C_\nu^{(\epsilon)}(0)} = \frac{(\tau+\epsilon)^\alpha - 2\tau^\alpha + |\tau-\epsilon|^\alpha}{2\epsilon^\alpha}. \quad (10)$$

To show that FBM is a viable approximation for the dynamics displayed by chromatin in the time range of our experimental measurements, we verified that displacements are self-similar Gaussian distributed with aforementioned covariance matrix and that its velocity autocorrelation agrees with theoretical predictions (Supplementary Figs. 7 and 8).

*Bayesian inference of diffusion parameters.* The GP provides the probability of observing a trajectory $r$ given $D_\alpha$ and $\alpha$. Then, we applied Bayes theorem[23] to obtain the posterior distribution over the diffusion parameters given the measured trajectory:

$$P(D_\alpha, \alpha, \mu|r) = \frac{P(r|D_\alpha, \alpha, \mu) P(D_\alpha, \alpha, \mu)}{\int P(r|D_\alpha, \alpha, \mu) P(D_\alpha, \alpha, \mu) dD_\alpha \, d\alpha \, d\mu}, \quad (11)$$

where $P(D_\alpha, \alpha, \mu)$ represents the prior distribution of the model parameters. Assuming a flat prior on $\mu$, $D_\alpha$ and $\alpha$, the log-posterior can be expressed as

$$\log(P(D_\alpha, \alpha, \mu|r)) \propto -\frac{1}{2}(r-\mu)^T \Sigma_{D_\alpha,\alpha}^{-1}(r-\mu) - \frac{1}{2}\log|\Sigma_{D_\alpha,\alpha}| - \frac{N}{2}\log(2\pi), \quad (12)$$

where N represents the number of points measured and $|\cdot|$ is the determinant function. To obtain maximum posterior estimates, we optimized (12) using the Nelder-Mead Simplex method[55]. In addition, we used the Metropolis-Hastings method[23,31] to sample the posterior probability distribution in order to calculate confidence intervals for our estimations. Note that, thanks to this Bayesian approach, available prior knowledge of the diffusion parameters can easily be incorporated into the analysis. For more information regarding MH sampler, view Supplementary Fig. 12.

*Incorporating of substrate movement in the GP-FBM framework.* In the main text, we introduced an extended GP-FBM model to deal with external sources of movement. Here, we present the derivation for two particles subject to a common substrate movement, however, it can be extended for an arbitrary number of particles using the marginalization rule of multi-variate Gaussian distributions[23]. The key idea is to assume that the movement of the particles with respect to the substrate as well as the movement of the substrate itself can be described by independent fractional Brownian motions. Therefore, the probability of observing the particle trajectories $a_1$ and $a_2$ with respect to a given frame of reference $R$ that moves with the substrate is:

$$\rho(a_1, a_2, R|\alpha, D_\alpha) \propto \exp\left( -\frac{1}{2}a_1^T\Sigma_1^{-1}a_1 - \frac{1}{2}a_2^T\Sigma_2^{-1}a_2 - \frac{1}{2}R^T\Sigma_R^{-1}R \right) \quad (13)$$

where $\Sigma_1$, $\Sigma_2$ and $\Sigma_R$ are FBM covariance matrices that are fully characterized given the diffusion parameters $D_\alpha = \{D_{\alpha,1}, D_{\alpha,2}, D_{\alpha,R}\}$ and $\alpha = \{\alpha_1, \alpha_2, \alpha_R\}$.

Next, to obtain the probability distribution over the trajectories $r_1$ and $r_2$, we applied the change of coordinates $r_i = a_i + R$ (see scheme in Fig. 2a) leading to the matrix expression,

$$\rho(r_1, r_2, R | \alpha, D_\alpha) \propto \exp\left(-\frac{1}{2}\begin{pmatrix} r_1 \\ r_2 \\ R \end{pmatrix}^T \begin{pmatrix} \Sigma_1^{-1} & 0 & -\Sigma_1^{-1} \\ 0 & \Sigma_2^{-1} & -\Sigma_2^{-1} \\ -\Sigma_1^{-1} & -\Sigma_2^{-1} & \Sigma_1^{-1} + \Sigma_2^{-1} + \Sigma_R^{-1} \end{pmatrix} \begin{pmatrix} r_1 \\ r_2 \\ R \end{pmatrix}\right),$$ (14)

Then, to marginalize the unobserved trajectory of the moving frame $R_i$, we need to calculate the inverse of the block matrix in equation (14). To do so, we use results from[57] on inverting a 2x2 block matrix such as

$$\Lambda = \begin{pmatrix} A & B \\ C & D \end{pmatrix}$$ (15)

according to the following result

$$\Lambda^{-1} = \begin{pmatrix} A^{-1} + A^{-1}B(D - CA^{-1}B)^{-1}CA^{-1} & -A^{-1}B(D - CA^{-1}B)^{-1} \\ -(D - CA^{-1}B)^{-1}CA^{-1} & (D - CA^{-1}B)^{-1} \end{pmatrix}.$$ (16)

Taking $A = [(\Sigma_1^{-1}, 0); (0, \Sigma_2^{-1})]$, $B = -[\Sigma_1^{-1}; \Sigma_2^{-1}]$, $C = B^T$ and $D = \Sigma_1^{-1} + \Sigma_2^{-1} + \Sigma_R^{-1}$, it is easily shown that the top-left corner of $\Lambda^{-1}$ is given by $[(\Sigma_1 + \Sigma_R, \Sigma_R); (\Sigma_R, \Sigma_2 + \Sigma_R)]$. Using this result, we can marginalize $R$ in equation (14) giving the expression,

$$\rho(r_1, r_2 | \alpha, D_\alpha) \propto \left| \begin{matrix} \Sigma_1 + \Sigma_R & \Sigma_R \\ \Sigma_R & \Sigma_2 + \Sigma_R \end{matrix} \right|^{-1} \exp\left(-\frac{1}{2}\begin{pmatrix} r_1 \\ r_2 \end{pmatrix}^T \begin{pmatrix} \Sigma_1 + \Sigma_R & \Sigma_R \\ \Sigma_R & \Sigma_2 + \Sigma_R \end{pmatrix}^{-1} \begin{pmatrix} r_1 \\ r_2 \end{pmatrix}\right).$$ (17)

This result clearly shows that the substrate movement induces a correlation between the particle trajectories as the off-diagonal elements of the block matrix in equation (17) are non zero. In addition, the covariance matrix of the substrate movement appears also in the diagonal terms, increasing the overall variance of the particles and their total movement. Consequently, if this correction is ignored the diffusion parameters are over-estimated.

*Inference of substrate movement.* Similarly as before, the diffusion parameters of the particles as well as the substrate can be estimated using equation (17) and Bayesian inference. We can also estimate how the substrate moves. For that, we can calculate the conditional distribution of $R$ given the particle trajectories as

$$\langle R \rangle = -\left(\Sigma_1^{-1} + \Sigma_2^{-1} + \Sigma_R^{-1}\right)^{-1} \left(\Sigma_1^{-1} \ \Sigma_2^{-1}\right) \begin{pmatrix} r_1 - \mu_1 \\ r_2 - \mu_2 \end{pmatrix}$$ (18)

with co-variance matrix given by $\Sigma_{\langle R \rangle} = \left(\Sigma_1^{-1} + \Sigma_2^{-1} + \Sigma_R^{-1}\right)^{-1}$.

Unfortunately, we could not find an analytical solution for this problem. Nonetheless, we can solve it numerically. In Supplementary Fig. 4.a we show an example of the estimated movement for the substrate with one standard deviation compared to the real simulated trajectory for a system with two particles. In Supplementary Fig. 4.b-c we display the overall accuracy when working with two or more particles.

**Benchmarking GP-FBM**

*Simulated trajectories.* We simulated single trajectories with 250 time points using the aforementioned Gaussian process with FBM kernel. To keep the benchmark as general and unbiased as possible, we uniformly sampled values for our parameters in the ranges $0.01 < D_\alpha < 1.5$, $0.01 < \alpha < 1.9$, $0.1 < dt < 1.0$ and $0.001 < \sigma < 0.25$. To benchmark a system of N particles affected by substrate movement, we generated N +1 trajectories and add the latter to all the others. Finally, a uniform distribution is used to remove 0% to 80% of points from each trajectory to simulate experimental occlusions.

*Mean squared displacement (MSD) implementation.* To calculate $D_\alpha$ and $\alpha$ for single trajectories, we estimate a MSD using a sliding window method. This method is mathematically defined as follows

$$\langle r_n^2 \rangle = \frac{1}{N-n}\sum_{i=1}^{N-n} \left(r_{i+n} - r_i\right)^2,$$ (19)

for a trajectory with N points and step interval $n$. Due to implicit correlations present in single trajectories, we use only initial 10% step intervals. To improve accuracy, we also estimate an average localization error $\sigma$. Finally, this experimental curve is approximated by the theoretical mean squared displacement equation

$$\langle r^2 \rangle = 4D_\alpha t^\alpha + 2\sigma^2,$$ (20)

from which diffusion parameters are inferred using linear regression. For more information[58].

*Displacement distribution based (DDB) implementation.* The theoretical expressions for the displacement distribution is obtained as a solution of the Fokker-Planck equation with localization error $\sigma$. In polar coordinates it takes the form

$$\rho(r, \theta | D_\alpha, \alpha, t, \sigma)\, dr d\theta = \frac{r}{2\pi(2D_\alpha t^\alpha + \sigma^2)} e^{-\frac{r^2}{4D_\alpha t^\alpha + 2\sigma^2}} dr d\theta.$$ (21)

In order to calculate experimental distributions for single cells, we resort to a sliding window method similar to the one present for MSD. Differently, we calculate normalized histograms with all the absolute displacement values. As before, we calculate an average localization error $\sigma$ to improve localization and use only histograms calculated for initial 10 step intervals. With these measurements, we optimize the equation above for $D_\alpha$ and $\alpha$ using Bayes approach with non-informative priors for both parameters.

**Law of total variance**. The law of total variance is used to determine how much of the measured variance comes from within or across samples. Starting off from the law of total expectation:

$$\langle x \rangle = \int dy\, \langle x | y \rangle \rho(y) = \langle \langle x | y \rangle \rangle,$$ (22)

we can calculate

$$\langle x^2 \rangle = \left\langle \text{var}(x|y) + \langle x|y \rangle^2 \right\rangle.$$ (23)

Subtracting $\langle \langle x|y \rangle \rangle^2$ from both sides

$$\langle x^2 \rangle - \langle x \rangle^2 = \left\langle \text{var}(x|y) \right\rangle + \left\langle \langle x|y \rangle^2 \right\rangle - \langle \langle x|y \rangle \rangle^2.$$ (24)

Upon algebraic manipulation, we obtain the final result

$$\text{var}(x) = \left\langle \text{var}(x|y) \right\rangle + \text{var}\left(\langle x|y \rangle\right),$$ (25)

which states that the total measured variance in $x$ is composed by the $\text{var}(x)$ given sample $y$ and $\langle x \rangle$ calculated for each $y$.

**Statistical analysis and reproducibility**. Data for interphase and mitotic cells are from two independent experiments, while Nocodazole-treated cells are from one experiment. Anchor data is accumulated from 11 independent experiments.

To compare diffusive properties or inter-probe distances across different loci or conditions (Fig. 3c, d, and 4b–d), we performed Wilcoxon rank sum tests. For inter-probe distances, the distributions of the median distances for each movie were used. For Fig. 4c, d, where fifteen pairwise comparisons are possible, the p-values were corrected for multiple testing with the Benjamini-Hochberg method and differences are considered statistically significant for p-values inferior to 0.05.

**Reporting summary**. Further information on research design is available in the Nature Research Reporting Summary linked to this article.

## Data availability
The data that support this study are available from the corresponding authors upon reasonable request. Microscopy data with analysis files that support the findings of this study have been deposited in Zenodo and can be accessed with https://doi.org/10.5281/zenodo.5359893[59], https://doi.org/10.5281/zenodo.5360028[60] and https://doi.org/10.5281/zenodo.5361054[61]. The source data are provided with this paper.

ES Hi-C sequence data from[42] were taken from Gene Expression Omnibus (GSE96107 [https://www.ncbi.nlm.nih.gov/geo/query/acc.cgi?acc=GSE96107]), and mapped to mm9 and normalized with FAN-C[62]. The normalized submatrix (chr6:51500000-53000000) was then extracted for visualization. ES and neuronal precursor cell H3K27ac and CTCF ChIP-seq data were taken as bigWig files from Gene Expression Omnibus; GSE96107 [https://www.ncbi.nlm.nih.gov/geo/query/acc.cgi?acc=GSE96107] for all except ES H3K27ac, taken from GSE49847 [https://www.ncbi.nlm.nih.gov/geo/query/acc.cgi?acc=GSE49847]) and visualized in R using the package rtracklayer. Source data are provided with this paper.

## Code availability
C++ libraries, batch templates and graphical user interface are available at https://github.com/guilmont[63].

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

## Acknowledgements

We thank Luca Giorgetti for providing the TetO ES line and for critical reading of the manuscript. This work was possible thanks to funding from grants by LabEx INRT

(ANR-10-LABX-0030-INRT, a French State fund managed by the Agence Nationale de la Recherche under the frame program Investissements d'Avenir ANR-10-IDEX-0002-02), CNRS << Osez l'interdisciplinarité ! >> , ERC (Starting Grant 678624 - CHROM-TOPOLOGY) and ATIP-Avenir. The microscopy was performed at the Imaging Center of the IGBMC.

## Author contributions

G.M.O. developed mathematical models, algorithms, and GP-Tool, also performed overall data analysis. D.K. generated anchor cell lines. A.O., D.K., and M.M. did experiments and validations. K.B. provided ANCHOR constructs and consulted on ANCHOR experiments. T.S and N.M. conceived, designed, and supervised the study. G.M.O., T.S., and N.M. wrote this manuscript with input from all other authors.

## Competing interests

The authors declare no competing interests.
