## [Peer Review File · Nature Communications]

Precise measurements of chromatin diffusion dynamics by modeling using Gaussian processesEditorial Note: This manuscript has been previously reviewed at another journal that is not operating a transparent peer review scheme. This document only contains reviewer comments and rebuttal letters for versions considered at Nature Communications

REVIEWER COMMENTS

Reviewer #1 (Remarks to the Author):

In this paper, Oliveira et al. develop a computational tool based on Gaussian Processes and Fractional Brownian Motion (GP-FBM) to estimate diffusion parameters of chromatin loci from their stochastic trajectories. Different from previous methods, GP-FBM takes into account higher-order correlations within the trajectories. It can also automatically take into account occlusions and localization imprecision, as well as substrate movement. By testing on thousands of in-silico simulated trajectories, GP-FBM is shown to outperform existing methods. GP-FBM is then applied to actual trajectories generated by live imaging of mouse ES cells to get insights into chromatin dynamics in different contexts. Interestingly, it is found that diffusion parameters are similar in interphase and mitosis despite the very different chromatin density. Instead, large variability is found across loci, which the authors suggest is linked to transcription.

I think that GP-FBM (and the developed graphical interface, GP-tool) is a valuable tool for the community to facilitate the study of diffusion dynamics of chromatin and other tracked particles. The study is sound, and the manuscript is well written and organized.

I have the following few remarks/suggestions:

- 1) It would be useful to specify how parameters that simulate occlusions and localization imprecision (10% of the points removed and positional noise of $1=10$ of a pixel) have been chosen. Are they based on the evaluation of experimental errors? It should be checked that the comparison of different methods is robust to the variation of such parameters.
- 2) The authors should explain how the contribution of inter-cell vs intra-cell variability (Fig.3e-f) has been evaluated. This result is interesting since, as stated in the paper, it suggests that variability may depend on specific chromatin or nuclear context more than on differences in the cell state. However, I couldn't find this information in the manuscript.
- 3) The tracking of three regions at the HoxA locus before and after induction of HoxA genes with retinoic acid helps to illustrate the possible applications of GP-FBM. However, it doesn't allow to conclude that there is a correlation of chromatin dynamics with transcription. The finding that region T1 is significantly more confined than T2 and T3 is not necessarily due to activity. After induction of the genes there are indeed changes in diffusion parameters, but they are different for each of the three regions and cannot be easily interpreted without further investigation. Thus, I would not say that there is a simple "correlation" with transcription.
- 4) The significance threshold used for p-values should be explicitly indicated. It would also help readability show significance in the plots (e.g., in Fig. 3c-d, 4c-e), maybe with an asterisk.
- 5) I would improve Fig. 5 about GP-tool by making it more informative about what the software can do.

Reviewer #2 (Remarks to the Author):

The authors present an analysis suite for anomalous diffusion of chromatin based on Bayesian methods. They demonstrate very impressive performance, even when the environment itself is moving and data in the sample trajectory are missing. This will certainly be an important and useful tool for the experimental and simulations-based study of chromatin dynamics. However, for the following reasons I cannot recommend publication of the present version in Nature Communications.

(1) The analysis is based on the fractional Brownian motion model. This is actually a simplification, as FBM describes motion whose driving noise is external, i.e., it is not defined by any temperature and Stokes-Einstein relation for the diffusion coefficient. The latter, D in the present paper, is a function of α and has dimension of $\text{cm}^2/\text{sec}^\alpha$. While these points should be mentioned, the role of the generalised Langevin equation discussed along with the generalised fluctuation-dissipation relation due to Kubo, as well as its relevance to living biological cells scrutinised, the critical point here is that FBM (as much as the "overdamped limit" of the fractional Langevin equation) has a very characteristic feature: the existence of anticorrelations. To qualify FBM as a relevant model the authors need to demonstrate the characteristic displacement autocorrelation of FBM and that, within error, the area under this autocorrelation vanishes. Otherwise the basic assumption of this analysis remains obscure.

(2) In this context note that possibly the first single particle tracking analysis in the nucleus of a mammalian cell, Phys Rev Lett 103, 018102 (2009), shows a distinct crossover from subdiffusion to normal diffusion, interpreted in the reptation model. No such crossover dynamics is discussed here. It would indeed be very surprising if a single, and quite simplistic, stochastic process could be used to explain such complex dynamics. This point needs thorough discussion.

(3) The second point that deserves thorough discussion is the Gaussianity of the data. Among other reports, Biophys J 112, 532 (2017) demonstrates highly non-Gaussian profiles of displacements along with an empirical exponential distribution of diffusivities. How Gaussian are the observed data here? This could be checked by the higher order moments advocated by the authors, e.g., in terms of the kurtosis or non-Gaussianity parameter. Such a discussion is vital for this work.

I also mention that the authors themselves discuss non-Gaussianity on the ensemble level when they talk about a range of diffusivities. In the sense of superstatistics, integrating a Gaussian over a (here constant) diffusivity distribution in an interval will produce a non-Gaussian distribution.

(4) I emphatically contest the use of "confinement parameter" for the anomalous diffusion exponent α . In fact, FBM is *free*, unconfined motion. The anomalous character of the diffusion enters through correlations. This has nothing to do with confinement in the sense of Mike Saxton's explanation of anomalous diffusion due to corrals. Using this term is seriously misleading the general reader, who is not familiar with the physical properties of anomalous stochastic processes.

(5) For Bayesian analysis using FBM and noisy FBM the paper Phys Chem Chem Phys 20, 29018 (2018) should be mentioned. In the Discussion deep learning approaches should be briefly mentioned, see, e.g., Biophys J 117, 185 (2019); New J Phys 22, 013010 (2020). What are the advantages of the Bayesian method? See also the following point.

(6) How is the likelihood (5) sampled concretely and what is the performance? Some information could be contained in the supplement. Moreover, is the sampling unique? What about other sampling methods?

(7) In contrast to what is stated at the beginning of the Results section, there is ample work in which particle diffusion is being characterised, just

see the large amount of papers on such topics within the last five years or so. In fact, in many contemporary measurements the probability density function of displacements is analysed, the distribution of the diffusion coefficient is calculated, or higher moments evaluated. See, e.g., the above-mentioned paper Biophys J 112, 532 (2017), or Phys Res Res 2, 022020(R) (2020).

(8) Could the analysis based on equation (1) be generalised? For instance, what about using a time dependent diffusion coefficient [similar to what is physically meaningful in systems with shape-changes or polymerisation---compare Phys Rev E 102, 012109 (2020); PNAS 107, 17864 (2020); Frontiers Phys 7, 124 (2019); Phys Rev Lett 126, 128101 (2021)]?

(9) In figure 1(a) I am puzzled by the seemingly periodic patterns of the dashed black line. Where do these come from? Please explain. In general, a more detailed explanation of what exactly one can see in this graph would be helpful.

(10) In figure 10(b) I guess what the authors mean is that they plot the probability density function of displacements, not the mean displacement? Please specify. Does it make sense that the distributions appear symmetric on a logarithmic time axis?

(11) In figure 10(f) the raw-corrected difference for the "Mito" data is huge, in fact, a factor of 2 or even more. How could this be? This needs a more thorough discussion.

In summary the current manuscript lacks a lot of detail. Several points need to be clarified before any publication decision can be taken.

We thank the reviewers for their positive and constructive feedback, and are pleased that both find GP-FBM a “valuable” (reviewer 1) and “useful” (reviewer 2) tool for the community. Reviewer 2 raised concerns about the validity of our model assumptions and a lack of some details. We have answered these fully (see point-by-point discussion below) in a revised version of the manuscript; we thank the reviewer, since we believe the manuscript is now strongly improved as a result of these additions.

Reviewer #1 (Remarks to the Author):

In this paper, Oliveira et al. develop a computational tool based on Gaussian Processes and Fractional Brownian Motion (GP-FBM) to estimate diffusion parameters of chromatin loci from their stochastic trajectories. Different from previous methods, GP-FBM takes into account higher-order correlations within the trajectories. It can also automatically take into account occlusions and localization imprecision, as well as substrate movement. By testing on thousands of in-silico simulated trajectories, GP-FBM is shown to outperform existing methods. GP-FBM is then applied to actual trajectories generated by live imaging of mouse ES cells to get insights into chromatin dynamics in different contexts. Interestingly, it is found that diffusion parameters are similar in interphase and mitosis despite the very different chromatin density. Instead, large variability is found across loci, which the authors suggest is linked to transcription.

I think that GP-FBM (and the developed graphical interface, GP-tool) is a valuable tool for the community to facilitate the study of diffusion dynamics of chromatin and other tracked particles. The study is sound, and the manuscript is well written and organized.

We thank the reviewer for such positive and encouraging comments.

I have the following few remarks/suggestions:

1) It would be useful to specify how parameters that simulate occlusions and localization imprecision (10% of the points removed and positional noise of $1=10$ of a pixel) have been chosen. Are they based on the evaluation of experimental errors? It should be checked that the comparison of different methods is robust to the variation of such parameters.

Yes, indeed the parameters were chosen to match the typical occlusion rate and localization errors observed in our experimental setup. We have clarified that point in the legend of Fig. 1a. To improve the benchmarking of GP-FBM against other methods, we have now performed more simulations in which we infer D_α and α upon an extended range of occlusion rates, localization error and time intervals between acquired images (now shown in revised Fig 1c). The relative robustness to the different parameters is further demonstrated in new supplementary Fig S1 and S2, showing better performance of GP-FBM across the whole range.

2) The authors should explain how the contribution of inter-cell vs intra-cell variability (Fig.3e-f) has been evaluated. This result is interesting since, as stated in the paper, it suggests that variability may depend on specific chromatin or nuclear context more than on differences in the cell state. However, I couldn't find this information in the manuscript.

To increase the clarity of the manuscript we have included a new subsection in the methods (“Law of total variance”) where we describe in detail how the inter-cell vs intra-cell variability is calculated. Briefly, we use the law of total variance which states that the variance of a random variable (i.e. D_α or

α) given a second random variable (i.e. the cell) can be decomposed algebraically into the component “explained” by the second random variable (i.e. the intercellular variability) and the “unexplained” component (i.e. the intracellular variability).

3) *The tracking of three regions at the HoxA locus before and after induction of HoxA genes with retinoic acid helps to illustrate the possible applications of GP-FBM. However, it doesn't allow to conclude that there is a correlation of chromatin dynamics with transcription. The finding that region T1 is significantly more confined than T2 and T3 is not necessarily due to activity. After induction of the genes there are indeed changes in diffusion parameters, but they are different for each of the three regions and cannot be easily interpreted without further investigation. Thus, I would not say that there is a simple “correlation” with transcription.*

We agree with the reviewer and have toned down the text throughout the manuscript saying that, in some cases, changes in diffusive properties correlate with altered histone modifications or CTCF binding. We find potential links to transcription very attractive, but providing sufficient mechanistic evidence is beyond the scope of this methods paper. Indeed, the most important result of this manuscript, which we believe is adequately supported, is that diffusive properties are heterogeneous, or in other words, locus- and context-dependent. Identifying the underlying “rules” and potential links to regulation of genome function will be an exciting avenue of research for the community, which will be facilitated by GP-FBM.

4) *The significance threshold used for p-values should be explicitly indicated. It would also help readability show significance in the plots (e.g., in Fig. 3c-d, 4c-e), maybe with an asterisk.*

To maintain transparency of the results, we have included all p-values (significant and non-significant) in the figures. We used the widely accepted convention for significance threshold of $p < 0.05$, which is now stated explicitly in the text (“Surprisingly, we observed no significant differences in the mean apparent diffusion or anomalous coefficients between interphase and mitotic chromosomes ($p \geq 0.05$)...”).

5) *I would improve Fig. 5 about GP-tool by making it more informative about what the software can do.*

We have modified Fig. 5 and its caption to provide more details about GP-Tool’s functionalities. However, it is difficult for one screenshot to show all of the software’s features in a non-confusing way. Therefore, we have now included the readme as a supplementary document, which contains an in-depth description of all the functionalities of GP-Tool, and detailed instructions on how to use them.

Reviewer #2 (Remarks to the Author):

The authors present an analysis suite for anomalous diffusion of chromatin based on Bayesian methods. They demonstrate very impressive performance, even when the environment itself is moving and data in the sample trajectory are missing. This will certainly be an important and useful tool for the experimental and simulations-based study of chromatin dynamics.

We thank the reviewer for the positive comments. We are glad the reviewer thinks that our method will be important and useful for the experimental community and found its performance impressive.

We have revised our manuscript in-depth and addressed the reviewer's important and useful comments.

However, for the following reasons I cannot recommend publication of the present version in Nature Communications.

(1) The analysis is based on the fractional Brownian motion model. This is actually a simplification, as FBM describes motion whose driving noise is external, i.e., it is not defined by any temperature and Stokes-Einstein relation for the diffusion coefficient. The latter, D in the present paper, is a function of α and has dimension of $\text{cm}^2/\text{sec}^{\alpha}$. While these points should be mentioned, the role of the generalised Langevin equation discussed along with the generalised fluctuation-dissipation relation due to Kubo, as well as its relevance to living biological cells scrutinised, the critical point here is that FBM (as much as the "overdamped limit" of the fractional Langevin equation) has a very characteristic feature: the existence of anticorrelations. To qualify FBM as a relevant model the authors need to demonstrate the characteristic displacement autocorrelation of FBM and that, within error, the area under this autocorrelation vanishes. Otherwise the basic assumption of this analysis remains obscure.

We thank the reviewer for raising this important point. Indeed, chromatin movement is a highly complex process, most accurately described at the moment by polymer models based on formal physical principles. However, these models require many parameters that are impossible to obtain from one experimental approach. This was not our goal here. Rather, we aimed to obtain accurate estimates of the apparent diffusion coefficient and anomalous exponent from experimentally measured trajectories, which could directly facilitate interpretation of cell (nuclear) biology studies. We have thus used FBM as an approximation to the more complex dynamics of chromatin. The reviewer is absolutely right that in the original version of the manuscript we did not provide sufficient evidence that measured trajectories present properties that are consistent with FBM, and consequently, that this model is a suitable approximation.

As mentioned by the reviewer here and in other points, the experimental trajectories require three properties to behave like FBM:

The displacement distributions for different time intervals must be Gaussian;

The displacement distributions for different time intervals must be self-similar (i.e. be identical after the proper scaling transformation);

The displacement/velocity autocorrelation must have a characteristic pattern, including a convergence to zero for longer period of times when $\alpha \leq 1$.

In the revised manuscript, we included extra information about FBM properties along with corresponding tests for all experimental results, showing self-similar displacement distributions that are consistent with a Gaussian (new supplementary figure S7) and agreement with theoretical predictions for the velocity autocorrelation (new supplementary figure S8).

In the main text we have included the following paragraph (lines 162-166): "Before applying the GP-FBM probabilistic framework, we first determined whether the measured stochastic trajectories can be well described, to a certain approximation, by self-similar Gaussian distributed displacements and it presents a certain velocity autocorrelation function, signatures of FBM (see Methods). Interestingly, that seems to be the case for chromatin movements at the time scale of this study...".

These results bolster the suitability of FBM as a phenomenological model to approximate chromatin dynamics, at least for the time-scales that are used in this study, and that are typically accessible with the experimental methods available.

(2) In this context note that possibly the first single particle tracking analysis in the nucleus of a mammalian cell, Phys Rev Lett 103, 018102 (2009), shows a distinct crossover from subdiffusion to normal diffusion, interpreted in the reptation model. No such crossover dynamics is discussed here. It would indeed be very surprising if a single, and quite simplistic, stochastic process could be used to explain such complex dynamics. This point needs thorough discussion.

We are aware that at different time scales chromatin show different diffusive regimes. In our case and due to the limitation of the experimental approaches, we can only assess chromatin dynamics for short time scales. Indeed, we recorded chromatin movement during two minutes at most. In fact, Phys Rev Lett 103, 018102 (2009) shows a similar range of diffusion parameters that we have found in our study for the aforementioned timescale. We have changed the text to clearly state the time scale at which we are measuring chromatin dynamics, and to explicitly say that our assumptions are applying over this timescale only. We also discuss our results in comparison with the one from Phys Rev Lett 103, 018102 (2009) (lines 246-248).

(3) The second point that deserves thorough discussion is the Gaussianity of the data. Among other reports, Biophys J 112, 532 (2017) demonstrates highly non-Gaussian profiles of displacements along with an empirical exponential distribution of diffusivities. How Gaussian are the observed data here? This could be checked by the higher order moments advocated by the authors, e.g., in terms of the kurtosis or non-Gaussianity parameter. Such a discussion is vital for this work. I also mention that the authors themselves discuss non-Gaussianity on the ensemble level when they talk about a range of diffusivities. In the sense of superstatistics, integrating a Gaussian over a (here constant) diffusivity distribution in an interval will produce a non-Gaussian distribution.

As described in our previous point discussing the suitability of FBM to approximate our experimental data, we now show Gaussianity (new supplemental figure S7). We also mention the possible limitations of our method in the Discussion (lines 244-250).

*(4) I emphatically contest the use of "confinement parameter" for the anomalous diffusion exponent alpha. In fact, FBM is *free*, unconfined motion. The anomalous character of the diffusion enters through correlations. This has nothing to do with confinement in the sense of Mike Saxton's explanation of anomalous diffusion due to corrals. Using this term is seriously misleading the general reader, who is not familiar with the physical properties of anomalous stochastic processes.*

Yes, we fully agree with the reviewer in this point. We intended to give an intuitive name for biologists that may not be familiar with anomalous diffusion to convey the idea that for $\alpha < 1$ the particle takes longer periods of time to explore the space than a traditional Brownian particle. However, we acknowledge that the term "confinement" leads to an important misunderstanding and decided to change it through the whole manuscript. We now refer to the parameter alpha as the anomalous coefficient. We thank the reviewer for pointing this out.

(5) For Bayesian analysis using FBM and noisy FBM the paper Phys Chem Chem Phys 20, 29018 (2018) should be mentioned. In the Discussion deep learning approaches should be briefly mentioned, see, e.g., Biophys J 117, 185 (2019); New J Phys 22, 013010 (2020). What are the advantages of the Bayesian method? See also the following point.

We thank the reviewer to point out to these interesting references and we have included them in the discussion (lines 260-269). This has helped us to better place our method in the context of the previous research in the field. Our Bayesian approach is very similar in spirit to the one presented in *Phys Chem Chem Phys* 20, 29018 (2018), but our implementation is fairly different. Our approach allows us to utilize the whole power of Gaussian Processes and incorporate straightforwardly estimated localization errors in each frame and to determine the most likely trajectory of any given particle, which can be used to infer particle's position at unobserved time points. More importantly, our extension to multiple-spots allows us to infer substrate movement and remove it from the analysis automatically. On the other hand, the machine learning methods, based on random forests and neural networks, are really attractive and original ideas, but the methods need to be trained with simulated trajectories with similar structure as the experimental ones and have to be preprocessed beforehand. In our case, we perform the fitting directly on the experimental trajectory without the need of preprocessing, hence we believe our method is more flexible to handle trajectories with different lengths or different levels of occlusion. Also the probabilistic nature of our methods makes it is very easy to incorporate localization errors in a per frame basis and infer spot positions in unobserved time points. Overall, although the machine learning methods are very interesting, we think that our approach brings new features that are worth considering.

(6) How is the likelihood (5) sampled concretely and what is the performance? Some information could be contained in the supplement. Moreover, is the sampling unique? What about other sampling methods?

We have now extended the description of the sampling in the methods ("Spot detection and tracking"; "Bayesian inference of diffusion parameters"). More importantly, we have added two new supplementary figures that show the sampling performance of the spot position with localization error (Fig. S10) and the sampling of the diffusion parameters (Fig. S12). Briefly, we have used a simple Metropolis-Hastings MCMC algorithm for sampling both the localization and the diffusion parameters. In Fig. S10 and S12 we show that the sampling converges fairly quickly in 10^3 - 10^4 iterations (taking ~10s with our computer setup, given in caption to Fig. S12), and to the same solution if we start in different initial points (Fig. S12). Based on this adequate performance, we did not need to explore more advanced sampling methods.

*(7) In contrast to what is stated at the beginning of the Results section, there is ample work in which particle diffusion is being characterized, just see the large amount of papers on such topics within the last five years or so. In fact, in many contemporary measurements the probability density function of displacements is analyzed, the distribution of the diffusion coefficient is calculated, or higher moments evaluated. See, e.g., the above-mentioned paper *Biophys J* 112, 532 (2017), or *Phys Res Res* 2, 022020(R) (2020).*

We apologize for this oversight. We have now changed the introduction to cite previous work more extensively, and thus place our method in a broader context (lines 31-33 and 39-45). We have also rephrased the beginning of the results section to emphasize that the novelty of our approach is to perform the analysis on the whole trajectory and thus including higher-order temporal correlations (lines 72-74) which, in the following sections, allowed us to interpolate trajectories given localization errors and correct for substrate movement. We believe this is the key element that enables the improvement in estimation accuracy compared to methods that analyze the probability density function of displacements.

(8) Could the analysis based on equation (1) be generalized? For instance, what about using a time dependent diffusion coefficient [similar to what is physically meaningful in systems with shape-changes or polymerization---compare Phys Rev E 102, 012109 (2020); PNAS 107, 17864 (2020); Frontiers Phys 7, 124 (2019); Phys Rev Lett 126, 128101 (2021)]?

We thank the reviewer for this very interesting perspective. This might be possible assuming that the Gaussianity is somehow preserved, as required by standard Gaussian processes. It could be troublesome to implement a kernel with generic time dependent diffusion and anomalous coefficients as it might lead to non-Gaussian displacements as the reviewer mentioned in point 3. In any case, the current version of GP-FBM allows for accurate estimation of apparent diffusion coefficient and anomalous exponent within the time constraints of commonly performed experiments, which was the main goal.

(9) In figure 1(a) I am puzzled by the seemingly periodic patterns of the dashed black line. Where do these come from? Please explain. In general, a more detailed explanation of what exactly one can see in this graph would be helpful.

The dashed black line in Fig. 1a is a stochastic trajectory sampled from a Gaussian Process with a FBM kernel. Given a finite set of times and the Cholesky decomposition of the resulting covariance matrix, we sample a trajectory from a multivariate Gaussian distribution. Indeed, as sub-diffusive particles take longer periods of time to explore space, we might expect a greater probability for two consecutive steps to be in opposite directions, which could create this resemblance to periodic patterns. In the same vein, $\alpha=1$ presents no preferred direction, while the particle has greater probability to have two consecutive steps in the same direction for $\alpha > 1$. Examples are present in the following image:

(10) In figure 10(b) I guess what the authors mean is that they plot the probability density function of displacements, not the mean displacement? Please specify. Does it make sense that the distributions appear symmetric on a logarithmic time axis?

We have now modified Fig10b to improve clarity. The figure is quite dense as we plot both the displacement distribution for certain time points and the mean displacement curve across time (in log-scale). On top of that, we compare the same experimental displacement distribution against the theoretical curves as expected for a model with static substrate (dashed lines) to the left and a model with moving substrate (continuous line) to the right. Hoping to improve understanding, we have included an inset to make the figure more understandable and modified the caption to improve the description.

(11) In figure 10(f) the raw-corrected difference for the "Mito" data is huge, in fact, a factor of 2 or even more. How could this be? This needs a more thorough discussion.

This is indeed an interesting observation. Mitotic embryonic stem cells tend to be more loosely attached to, or can even temporarily detach from their colonies, and hence are more mobile during the process of mitosis. Thus the substrate movement correction may have a larger impact in the estimation of the diffusion parameters for these cells. On the other hand, cells experimentally arrested in mitosis by nocodazole are allowed to sediment onto the surface of the glass before imaging and that may be the reason why the substrate movement correction is less important in this case. We have added these comments in the text as a possible explanation for the large correction factor observed in mitotic cells.

In summary the current manuscript lacks a lot of detail. Several points need to be clarified before any publication decision can be taken.

We are grateful for the reviewer's constructive criticism as we believe it helped us to improve the manuscript greatly. Most significantly, we have now demonstrated that FBM can be used to approximate chromatin dynamics within the timescales of our experimental setup and, hopefully, cleared up some confusing aspects of the manuscript, particularly those pertaining to the definition of the anomalous exponent along with GP-Tool robustness and sampling rapid convergence.

REVIEWERS' COMMENTS

Reviewer #1 (Remarks to the Author):

The authors have addressed my previous concerns and have clarified some aspects not well described previously. In addition, their new analyses have significantly strengthened the manuscript, which I think is suitable for publication.

Reviewer #2 (Remarks to the Author):

I thank the authors for their very careful revision. They managed to successfully counter all my criticism. The new version reports important and timely results of a level that is fully adequate for the high profile venue of Nat Comm. I very warmly support publication.